# COVID-19 and Sleep Disturbances: A Literature Review of Clinical Evidence

**DOI:** 10.3390/medicina59050818

**Published:** 2023-04-22

**Authors:** Likhita Shaik, Sydney Boike, Kannan Ramar, Shyam Subramanian, Salim Surani

**Affiliations:** 1Department of Family Medicine, Hennepin Healthcare, Minneapolis, MN 55415, USA; 2Department of Medicine, University of Minnesota Medical School, Minneapolis, MN 55455, USA; 3Mayo Clinic, Rochester, MN 55905, USA; 4Sutter Gould Health System, Fresno, CA 93720, USA; 5Department of Pulmonary, Critical Care and Sleep Medicine, Texas A&M University, College Station, TX 77843, USA

**Keywords:** sleep disorders, post-COVID-19 syndrome, coronasomnia, sleep cycle, COVID-19, pandemic

## Abstract

The need for adequate good quality sleep to optimally function is well known. Over years, various physical, psychological, biological, and social factors have been investigated to understand their impact on sleep. However, understanding the etiological processes that are involved in causing sleep disturbances (SD) as impacted by stressful phases such as pandemics has not been well studied. Many such etiological and management strategies have surfaced during the latest “coronavirus disease of 2019 (COVID-19) pandemic. The occurrence of these SD in the infected and uninfected individuals poses a need to investigate factors linked to such occurrence during this phase. Some of such factors include stressful practices such as social distancing, masking, vaccines, and medications availability, changes in routines, and lifestyles. As the status of infection improved, a collective term for all the prolonged effects of COVID-19 after the resolution of the primary infection called the post-COVID-19 syndrome (PCS) surfaced. Apart from impacting sleep during the infectious phase, the aftereffects of this virus left an even greater impact during the PCS. Various mechanisms have been hypothesized to be linked to such SD during the PCS, but the available data are inconclusive. Further, the varied patterns of incidence of these SDs differed by many factors, such as age, gender, and geographical location, making clinical management even more challenging. This review elucidates the impact of coronavirus 2 (SARS-CoV-2) (COVID) disease on sleep health during the various phases of the COVID-19 pandemic. We also investigate different causal relationships, management strategies, and knowledge gaps related to SD during the COVID-19 pandemic.

## 1. Introduction

The initial outbreak of “coronavirus disease of 2019. (COVID-19), which would cause a devastating pandemic worldwide, occurred in late 2019. On March 11th, 2020, the World Health Organization (WHO) announced the COVID-19 global pandemic outbreak [1]. The World Health Organization has reported that this pandemic has surpassed the number of deaths due to respiratory diseases from prior years, amounting to nearly 6.5 million deaths worldwide [2]. The incubation period of 5–14 days made it difficult to recognize infected individuals during the asymptomatic phase of the infection, making it challenging to isolate patients while they are contagious [3], further complicating the management of the disease. The underestimation of the effects of this new virus aided in its further spread and provided an unguarded environment for multiplying the contagious state exponentially, making the task of virus containment extremely challenging. The increased morbidity and mortality associated with the virus speeded actions toward curbing the further spread [4]. These included track-and-tracing, self-isolation, quarantine, social distancing, and community containment, and nationwide lockdowns were among the public health measures to prevent the further spread [5]. The sudden climb in awareness regarding the gravity of the situation created monumental havoc among the masses. In addition to facing the direct effects of COVID-19 infection, people’s health was additionally affected by the guidelines put in place in an attempt to slow the spread. Many perceived isolation as confinement, causing worries about livelihood and social distancing from loved ones, directly and indirectly affecting the health and well-being of patients and their families [6]. The additional media updates regularly made on the statistics of COVID-19 significantly affected mental health and sleep, causing distress among many [7]. 

Sleep is vital for the optimal functioning of organ systems and the mind. Poor sleep is associated with unfavorable clinical outcomes. Such outcomes are undesirable at any point in time, especially amidst a raging pandemic [8]. Research studies indicate the link between infectious disease outbreaks and sleep dysfunction [9,10,11,12]. Similarly, associations between infectious outbreaks and sleep dysfunction can be applied during this COVID-19 pandemic in the setting of physical illness, separation from family and friends, environmental stresses, social isolation, and prior poor mental health [13,14,15,16,17]. Jaharmi et al., describe sleep changes as a common symptom among COVID-19-infected individuals [8]. As per a meta-analysis, the global prevalence of COVID-19-related sleep disturbances was 40.49% [18]. However, it remains a scarcely studied entity in the realm of COVID-19 research [19]. The psychological or psychiatric repercussions of such measures have been well-researched. However, the effects on sleep are not well understood due to the lack of consistency between study results [20,21,22]. We discuss these differences further and provide insights about the sleep disturbances related to the COVID-19 pandemic.

## 2. Sleep Disturbances: Definition and Instruments

Sleep disturbances refer to a group of disturbances described as trouble falling/staying asleep, which can cause excessive drowsiness during the daytime as a result of an inadequate amount of sleep, or change in sleep quantity, quality, or timing [23,24,25]. However, the sleep disturbances that were focused on in the majority of the studies were poor sleep quality and insomnia. Sleep quality that was poor in nature was assessed using the Pittsburgh sleep quality index (PSQI), and insomnia was assessed using the insomnia severity index (ISI) and the Athens insomnia scale (AIS). Many other measures were also used to report sleep disturbances. The BEARS questionnaire described bedtime problems, increased daytime sleepiness, awakening throughout the night, regularity and duration of sleep, and finally snoring. Adequate sleep, somnolence, amount of sleep obtained, snoring, and waking up with shortness of breath or with a headache were reported using the medical outcomes study sleep scale (MOS-SS). Patient’s own perceptions of sleep quality, the depth of their sleep, and restoration associated with sleep were assessed by the patient-reported outcomes measurement information system-sleep disturbance (PROMIS) questionnaire. In children, the Sleep Disturbance Scale for Children (SDSC) and the youth self-rating insomnia scale were valuable tools to measure these sleep changes [18].

## 3. Prevalence of Sleep Disturbances in COVID-19 Patients

Sleep’s vital role in everyday life is well known [23]. Sleep dysfunction has been linked to deleterious effects on mood, health, and various motor vehicle accidents [24]. Self-reported data from various sections of society indicate high amounts of the burden of this dysfunction during the COVID-19 pandemic [25]. The prevalence of sleep changes during the COVID-19 pandemic was estimated to be about 36% [8,25], with higher impact on patients infected with the COVID-19 virus [8,25,26,27,28]. However, these numbers are variable based on several factors, as discussed below. 

Age and gender: Among COVID-19 patients, the elderly and male gender exhibited a higher prevalence of sleep problems [8,25]. The study conducted in 51,625 participants indicated a 29% worsening of sleep changes (subjective deterioration in sleep quality) in individuals > 50 years of age during the COVID pandemic with a greater predilection toward the male gender. Their questionnaire also assessed sleep changes to “more or less trouble sleeping since the outbreak,” and this was recoded as a binary variable to compare “more trouble sleeping since the outbreak” versus “less trouble sleeping since the outbreak” or “no changes in sleep”. Participants who felt lonelier since after the COVID outbreak had more trouble sleeping than those who did not complain of loneliness (Odds Ratio (OR)  =  1.21; *p*  =  0.002; CI, 1.07–1.37 and OR  =  4.06; *p*  =  0.000; CI, 2.75–5.99, respectively). They also concluded that age and gender were independent predictors of sleep changes in their population [29].

Studies by Ara et al. [30], Wang et al. [31], Czeisler et al. [32] and Kim et al. [33] reported that sleep disturbances are either more likely among patients more than 30 years of age or increase with age. On the contrary, studies by Beck et al. [34], Romero-Blanco et al. [35], Mareli et al. [36], and Kaparounaki et al. [37] indicate a higher prevalence of sleep disturbances among younger ages (18–34 years). It can be argued that even though sleep structure and circadian rhythm alter with age, the variability in stress related to financial, social, and emotional burdens heavily impacts various age groups differently [17]. Moreover, as the above factors change with geographical location, cultural backgrounds and other comorbidities, inconsistency in results is expected [17,18]. 

Similar differences in sleep disturbances between males and females varies with various factors such as immunologic response to COVID-19 infection, underlying brain structural differences across sexes, and stresses related to working patterns [14]. The literature indicates that owing to the divergent emotional processing, women are more likely to report psychological comorbidities during mentally taxing situations [17]. This can develop inconsistencies in SDDCP. Thus, a few studies [30,38,39,40,41,42,43] show female gender preponderance to SDDCP, which is in contrary to studies indicating male preponderance [44].

However, further sub-analysis by Jaharami et al. concluded that female participants had higher prevalence rates of sleep disturbances, which is in concordance with prior studies [18].

Special populations: COVID-19-infected individuals had the highest rates of sleep disturbances at 52.39%, followed by 45.96% of children and adolescents, 42.47% of healthcare workers (HCWs), 41.50% of special populations with healthcare needs, 41.16% of university students, and 36.73% of the general population. Elderly HCWs and female university students were at an increased risk of sleep disturbances during this COVID-19 era [18]. Due to the highly stressful and demanding work and irregular shift schedules during the pandemic, HCWs (physicians, nurses, emergency medical personnel, dental professionals, diagnostics professionals, pharmacists, and administrative staff) suffered severe sleep changes such as poor sleep quality and insomnia, especially for the frontline workers [8,25,45,46,47,48]. That being said, there were not any significant differences among the types of (i.e., frontline vs. nursing) [18]. The prevalence of poor sleep quality (PSQI global score of >5) was high among pregnant patients, with an observed increase in prevalence with the advancement of gestational age and maternal age. It ranged between 59.5% in Indonesian studies and 88% in Turkish studies [49]. In a meta-analysis, these numbers were reduced to a pooled prevalence of 13% [50].Geographical region and Ethnicity: Latin America and the Caribbean populations reported poorer sleep than Europe, Central Asia, and North America [51]. However, a metanalysis by Scarpelli in 2022 reported poor sleep quality and higher insomnia scores most prominently in America [52]. The Black populations, more than White populations, reported symptoms of insomnia during the pandemic. However, this was no longer significant once there was adjusting for the Coronavirus Impact Scale scores [53]. Regarding sleep duration, Black respondents reported sleeping approximately one less hour of sleep compared to American Indian/Alaskan Native (AIAN), Asian, White, and Latinx young adults. Mediation analyses showed that disparities in duration of sleep between Asian and Black young adults might be explained by the increased likelihood of Black participants operating as workers deemed essential. Regarding quality of sleep, Black respondents were noted to have lower levels than Latinx, White, AIAN, and Asian young adults. Including coronavirus victimization distress as a factor in sleep quality resulted in a decrease in differences between Asian and White respondents, which suggests that coronavirus victimization distress somewhat explains differences between Blacks and Asians and Black and White young adults [54].

The metanalysis from various studies concludes that prevalence rates show differences in the sleep disturbances burden and were not affected by factors such as age, gender, and country [25].

## 4. How Did COVID-19 Affect Sleep: Related Mechanisms

The impact of the COVID-19 pandemic has affected sleep in various ways through cytokine release, neuro-inflammation, and other ways discussed later in this review. The psycho-sociological effects of social distancing, isolation, and quarantine measures manifested as loneliness, detachment, boredom, and loss of freedom [7,13,14,15,16,17]. This diminished coping mechanisms, which likely contributed to new mental health issues. As we already know, any amount of mental health concerns is an obvious link to sleep disturbances and vice-versa [7,13,14,15,16,17]. The increased morbidity and mortality in society further aggravated stress and led to sleep disorders. The direct and indirect mechanisms due to COVID-19 disease that are involved in causing sleep disturbances are discussed below [7,13,14,15,16,17].

### 4.1. Life Changes: Social Distancing, Isolation, Lockdown Measures, Economy Changes

The COVID-19 pandemic has caused huge amounts of psychological distress and poor mental health [13,55,56,57,58]. Research demonstrated that the prevalence of stress, depression, and anxiety during the COVID-19 pandemic is 29.6%, 33.7%, and 31.9%, respectively [59]. As these rates were comparable to the prevalence of sleep dysfunction, an obvious correlation between the two is assumed. Fear around COVID-19 was elevated by quarantine, isolation, and lockdown measures. The imposition of social distancing, masking, frequent hand hygiene, and isolation as initiatives to minimize the spread was also distressing. Studies reported an increased amount of sleep disturbances during the lockdown as compared to those without lockdown (~38% vs. ~43%). However, these studies were not statistically significant during the early pandemic until the full-fledged pandemic set in (36% vs. 47%) [18]. Stricter quarantine measures were associated with poorer sleep [51]. Life changes such as unemployment, difficulties transitioning to working from home, and increased domestic chaos in families led to poorer sleep patterns during COVID [60,61,62,63,64]. Imposing travel restrictions made it difficult for people to visit loved ones, which further amplified stress [65]. This situation worsened when social media and news channels reported the rampant death rates [66]. Many reported delayed sleep–wake cycles due to such changes [51]. Studies demonstrated that factors such as older populations, having a family member with COVID-19 infection, symptom changes after hospitalization, the number of current symptoms, and poor prior sleep quality affected their stress levels and further worsened sleep quality [9,25,67]. Lockdown caused the shutdown of schools, educational institutes, entertainment, and activity areas, ruining normalcy for children and adolescents [68]. The nature and scope of impact in this age group, as stated by Singh et al., were based on many vulnerable factors such as their developmental age, current status of education, having special needs, mental health conditions that existed prior to the pandemic, economically underprivileged, and child/parent(s) being quarantined due to fear of infection or active infection [69]. Evidence from research showed that children experienced disturbed sleep, experienced nightmares, appetite that was poor, agitation and inattention, and finally separation-related anxiety during the COVID-19 pandemic [70,71]. Moreover, the perception of the patient’s disease state by the patient was a clear indicator of mental distress and sleep dysfunction [12,18]. The elderly, those who have a partner, and those who live in a country with higher income were associated with better sleep [18,53,72]. However, few studies demonstrated delayed sleep–wake cycle changes due to work-from-home schedules. Individuals started to spend more time in bed, thus altering wake–sleep times, sleep duration changes, sleep disturbances, and paradoxically poorer sleep quality [51]. This emphasizes the importance of a healthy sleep routine during lockdown times as well. From all these studies, it is clear how social factors revolving around COVID play a role in the cycle of sleep distress.

### 4.2. Immunomodulation Driving Poor Sleep

Various research studies have demonstrated neurological symptoms as early signs of the infectious state. Sleep changes are the most frequent of these neurological signs, followed by headache, loss of taste, smell, and low mood [72]. Among the sleep variations during a COVID-19 infection, insomnia is the most frequent, making up approximately 80% of the differences seen in total sleep disturbances [18]. These symptoms had more predilection toward the female gender, likely attributable to the gender-based immune response variations [73]. Among the hospitalized COVID-19 patients that had neurological complications related to the infection, ~66% reported having sleep disturbances, fatigue, depression, and anxiety. Patients that required invasive ventilation through intubation were known to be associated with a poorer sleep quality. These changes can be directly linked to the neuroinflammation inflicted by the COVID-19 virus. Among the factors that influence this state of social isolation are the viral infection’s severity, patient immune system reactions, different treatments applied as corticosteroids, the Intensive Care Unit (ICU) stay, and the stigma enforced socially [74,75]. Studies indicate that duration of hospital stay, mental health concerns that existed prior to the pandemic, lower absolute lymphocyte count, and a neutrophil-to-lymphocyte ratio that is increased have been associated with a higher risk of dysfunctional sleep among COVID-19 patients [45]. As is well known, COVID-19 causes an inflammatory storm that causes the release of various cytokines and chemokines such as interleukins, TNF, granulocyte colony-stimulating factor (G-CSF), monocyte chemoattractant protein-1, macrophage inflammatory protein 1 alpha, C-reactive protein, ferritin, and D-dimers in blood. These immunomodulators, when entering the brain or spinal cord, can induce circadian rhythm changes leading to circadian rhythm disorders. This impact can be disproportionate to the immunomodulators released in COVID-19 [76,77,78] and hence causes a deleterious impact on sleep.

### 4.3. Post-COVID Syndrome (PCS)

After the resolution of the primary COVID-19 infection, most patients experience a prolonged myriad of symptoms, termed post-COVID-19 syndrome. Among other symptoms such as fatigue, muscle weakness, anxiety, and depression, sleep dysfunction is a frequent complaint under the umbrella of this syndrome [79], as demonstrated by El Sayed et al. and Islam et al. [80,81]. Mazza et al. reported a prevalence of post-traumatic stress disorder, depression, anxiety, and insomnia in 28%, 31%, 42%, and 40%, respectively, of the post-COVID-19 survivors after a four-week follow-up [24]. Studies report poor sleep quality after COVID-19 infection, demonstrated by a high PSQI score. There was a noted increase of 33% in the prevalence of insomnia after COVID-19 before the infection [19]. The abnormal values on the Insomnia Severity indices during the post-COVID-19 syndrome phase pointed toward an increase in insomnia symptoms that significantly reduced the quality of life for the affected individuals [82]. As discussed earlier, the existence of a strong interdependent relationship between mental health and the development of sleep dysfunction plays a vital role in insomnia evolution during the post-COVID-19 phase [19,83,84,85].

## 5. Various Sleep Disorders Associated with COVID-19 Infection

Prior epidemics and pandemics have exposed the world population to stress, social confinement, limited daylight exposure, step back from normalcy, and induced sleep disturbances, which during the coronavirus period was termed the coronasomnia phenomenon. These sleep disturbances include insufficient sleep, insomnia, poor sleep quality, sleep apnea, and circadian rhythm disorders [86]. Confinement from quarantine measures played a vital role in causing insomnia, as evidenced by studies in Europe [6]. In Italy, anxiety from COVID-19 was associated with sleep problems [41,42,87]. REM sleep disorders could be directly linked to the COVID-19 infection itself [88].

The prevalence of insomnia was as high as 60% among COVID-19 survivors, although variable rates were noted in China (2%) [89], Italy (40%) [90], and Mexico (77%) [90,91]. These results are attributable to the neuroinflammatory changes or the stress around the pandemic. These results may be related to the duration of illness, as explained by Pedrozo-Pop et al., where insomnia was more likely in patients who experienced illness for more than 3 weeks. Moreover, insomnia was associated with increased post-traumatic stress disorder (PTSD) risk as encountered in clinical practice [89,92,93].

Obstructive sleep apnea (OSA) was present in 8–21% of patients hospitalized with COVID-19 infection [46,94,95,96,97,98]. The results of the association between the two have been contradictory. However, the majority of studies suggest OSA as a risk factor for a complicated hospital course. The underlying inflammatory state of COVID provides a plausible explanation for this association. The hypoxemic state in OSA affects the cytokine storm, further worsening it [89]. Hence, owing to its antioxidant, anti-inflammatory, and pro-immunity actions, melatonin can be used in the treatment of COVID-19 pneumonia, especially in OSA patients [99,100]. Various studies thus indicate the relationship between COVID and OSA as bidirectional and synergistic [46,101,102]. However, minimal literature is available confirming this relationship and the prevalence of other sleep disorders during the COVID pandemic.

## 6. Poor Sleep and COVID-19

Sleep is a known factor that impacts the immune response, and evidence suggests that sleep disturbances alter the immunity to feasibly cause immunosuppression [103,104]. This may be from the changes in cytokine production, release, and associated cell count variability. Similarly, as studied by previous research, the COVID-19 virus alters sleep quality in hospitalized patients. However, the effect from the direct impact of the virus is ambiguous. It is hypothesized to be a result of anxiety, pain, physical discomfort, prolonged hospitalization, or polypharmacy [105,106]. Several studies indicate the link between poor sleep quality and slow improvement of lymphopenia in hospitalized patients and further need for ICU admissions, indicating the association between the two [107]. Research indicates that reduced sleep duration and coexisting comorbidities are significant indicators of disease severity. Hence, while studying the impact of COVID-19 on sleep, it should be borne in mind that sleep affects COVID-19 severity, alters the clinical course of events, and impacts the clinical outcomes of COVID-19. Outpatient and hospitalized patients suffering from this infection should be notified about the importance of sleep in the recovery from infection [108]. A particular study showed poor sleep and high-stress levels in hospitalized schizophrenic patients suspected of developing a COVID-19 infection. This indicates that treating sleep problems is imperative for recovering from mental illnesses and possibly strengthening immunity to prevent infections [109].

Sleep manifestations such as insomnia can impose prolonged problems. Insomnia can become chronic, worsen pain syndromes and gastrointestinal disorders, and increase the risk of hypertension and heart diseases if left untreated [28]. Recent evidence suggests that sleep loss is a potential biomarker of blood–brain barrier (BBB) leak. Possible mechanisms implicated in this process are neuroinflammation from causing loss of BBB integrity [110] and possibly inducing infection or other mental illnesses. Sleep stands as an often-ignored entity during the other deleterious impact of COVID-19 infection. However, it remains a silent component that can cause long-term effects, making early management vital to prevent such an outcome.

## 7. Management Modalities

Sleep is affected by and, in turn, also affects health and mood. Hence, it is imperative to recognize and address sleep disturbances in various population groups during the early phase of impact. This starts with identifying individual, interpersonal, professional, and community risk factors [111]. Basic principles of sleep hygiene should be reinforced. As COVID-19 infection has been correlated to sleep disturbances, mental health problems, and stress in general, special focus on this subset of patients is much needed to reduce the burden of the already disastrous disease impact and improve prognosis [16,99,106]. Often, hospitalization causes delirium and can disturb the sleep cycle. Hence, delirium management strategies such as frequent re-orientation, maintaining the sleep–wake cycle through adequate light settings, adequate pain control, and through use of circadian rhythm-maintaining medications such as melatonin [99,100]. The effects of ICU stay and associated trauma from mechanical ventilation can induce PTSD and aggravate sleep problems [89,90,91,92,93]. The use of prophylactic melatonin and melatonin receptor agonists such as Ramelteon for sleep–wake cycle disorders improved the sleep of ICU patients, decreasing the delirium occurrence and duration of ICU stay [100]. Other sleep dysfunction treatment modalities have been discussed below (Figure 1).

## 8. Novel vs. Existing Therapeutics

This work has important biomedical applications, as it provides a review on the literature surrounding actions providers can take in treating sleep disturbances. Some of these examples, as previously mentioned, include good sleep hygiene, strategies of minimizing delirium in the inpatient setting, and utilizing melatonin agonists when indicated. Other nonpharmacologic treatments classically discussed in the sleep literature include sleep diaries, sleep restriction, and other cognitive and relaxing techniques (122). Other pharmacologic therapies include benzodiazepines, non-benzodiazepine hypnotics, antihistamines, and antidepressants (122). Comparatively, our therapies discussed offer many benefits, including ease of administration (i.e., minimizing delirium via maintaining schedules, opening window blinds) and accessibility to patients (i.e., sleep hygiene is a factor patients can directly address themself). More research needs to be conducted regarding use of these alternate therapies in sleep disturbances.

## 9. Barriers to Significance of Available Evidence

Much research conducted in the field to link sleep pattern changes during the COVID era to the effects of COVID infection itself or the psychological changes pertinent to COVID infection may be confounded by various variables. As described above, lockdown has been a significant impacting factor of these sleep changes. However, the intensity of lockdown was variable in different countries and so is the impact on sleep factors and ultimately sleep [47]. Moreover, it is difficult to quantify the intensity of sleep disease burden due to COVID without eliminating other confounding variables. Among these factors, existing sleep disorders, mental health disorders, and other relevant personal medical history that can impact sleep or mental health play a huge role. Other variables include gender, occupational status, age, medical conditions, lifestyle changes involving health behavior (eg, sedentary behavior, maladaptive eating habits) [8,17,112]. For example, Pataka et al. described comorbidities such as obesity, hypertension, diabetes, and cardiovascular disease as strong predictors of worse clinical outcome in COVID-19 patients and could act as confounding factors in the development of OSA [112]. Partinen et al. discuss daytime sleepiness changes from ethnic, cultural, and meteorological differences [113]. Many such confounders need to be eliminated to generalize data and improve the clinical significance of current evidence.

## 10. Future Research

Melatonin and melatonin agonists have been hypothesized to be useful as an antioxidant in acute respiratory distress syndrome (ARDS)-related acute lung injury and ventilator-associated acute lung injury. However, more studies are required to support these hypotheses in different populations [114,115,116].Most of the research on sleep changes in COVID is related to the effect of psychosocial stressors. The direct effects of the coronavirus itself on sleep have not been investigated yet. This area of research would be useful to complete knowledge gaps to explain sleep disorders in patients with minimal stress or good mental health.CRP and hs-CRP have been hypothesized to be indicators of OSA [117,118,119]. This effect needs to be further investigated.Limited evidence is available on the variability of the different types of coronaviruses on the blood–brain barrier (BBB). More research in this area will provide better insight into the variability of the difference in sleep dysfunction variation among the different coronavirus types.Newer technologies tag melatonin with components such as selenium, Chitosan, solid lipid and nanostructured lipid carriers, Liposomes, poly (d,l-lactide-co-glycolide) (PLGA), or poly (ethylenglycole) (PEG) to form nanoparticles. These nanoparticles are used as diagnostic methodologies and in vivo melatonin delivery systems. Despite the increasing uses of melatonin, lack of evidence in human subjects paves path for further research in this area [87,120,121] (Figure 2).

## 11. Conclusions

The COVID-19 pandemic has resulted in various sleep disturbances, among which insomnia has been the most common. These disturbances have majorly impacted COVID-19 infected patients, children and adolescents. Lockdown and prior mental health problems proved detrimental to sleep and have been key contributors to these sleep changes. Further research is needed to focus on eliminating potential confounding variables to answer if these sleep changes are truly associated with COVID-19 infection itself. There is a need to investigate the role of alternate therapies in the management of these sleep disturbances in various population groups with varied disease spectra.

## Figures and Tables

**Figure 1 medicina-59-00818-f001:**
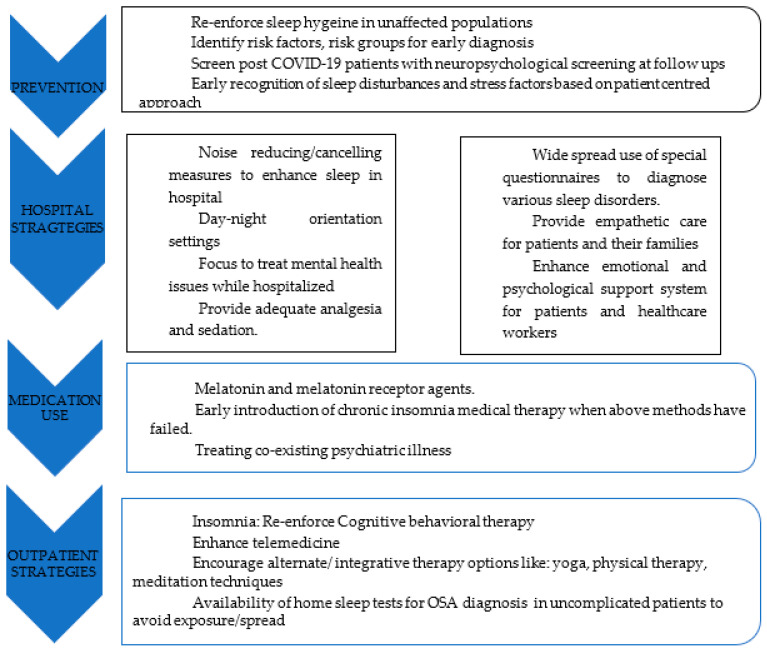
Treatment modalities for pandemic related sleep dysfunction.

**Figure 2 medicina-59-00818-f002:**
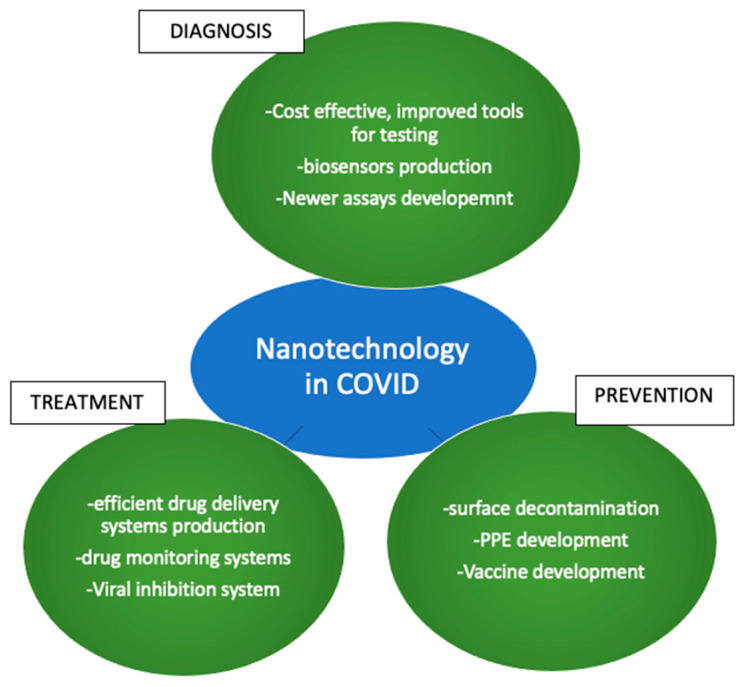
Uses of nanotechnology in COVID-19 pandemic.

## Data Availability

Data available in a publicly accessible repository that does not issue DOIs Publicly available datasets were analyzed in this study. This data can be found here https://pubmed.ncbi.nlm.nih.gov/.

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
