# Peer review of "COVID-19 and Sleep Disturbances: A Literature Review of Clinical Evidence"

_medicina, 2023, doi:10.3390/medicina59050818_

Round 1

Reviewer 1 Report

Authors have presented an overview about the sleep disturbances related to the COVID-19. The topic is relevant especially under current circumstances of Post- and Long-COVID, and the authors here appropriately indicated the unexplored areas of research that should be addressed in future studies. Briefly, authors highlight the high prevalence of sleep problems among COVID-19 infected patients, children and adolescents, and lockdown and prior mental health problems proved detrimental to sleep and has been a key contributor to the reported sleep changes. Finally, authors have emphasized the importance of future studies to address the confounding factors such as gender, occupational status, age, medical conditions including but not limited to sleep and mental health disorders, and investigation of direct causal effects of the Coronavirus on sleep, ultimately to generalize available data and improve the clinical significance of current evidence.
I do not have any major concern, however, there are few minor suggestions, and addressing those will help improve the article.
1. More than 11-12 citations have been repeated at least twice in the reference list. This should be corrected with proper referencing tool.
2. At line 63-66, Authors have indicated about the lack of consistency between study results and to discuss these differences as one of the goals of this review. However, except few places such as at line 180, it is not apparent elsewhere in the current manuscript. Please briefly discuss or summarize those including the information based upon the listed citations 20-22.
3. The manuscript could greatly benefit from additional and updated information from few of the recent publications. The latest citation used in this manuscript is almost a year old (April-May 2022). Few of relevant recent publications are listed below for authors consideration:
1. Scarpelli S et. al., 2022. Subjective sleep alterations in healthy subjects worldwide during COVID-19 pandemic: A systematic review, meta-analysis and meta-regression, Sleep Medicine, 100, 89-102, https://doi.org/10.1016/j.sleep.2022.07.012.
Discusses- Sleep quality in different world areas, characterization of subjective sleep alterations during the COVID-19 pandemic, identification of significant predictors of sleep problems:
2.Bezner TL, Sivaraman M. COVID-19 Pandemic and Its Impact on Sleep Health: A Rapid Review. Mo Med. 2022 Jul-Aug;119(4):385-389. PMID: 36118813; PMCID: PMC9462915.
Discusses- Direct Impact of COVID on Sleep Health and the Unfortunate Synergy with OSA
3. Schilling C et. al., Cognitive disorders and sleep disturbances in long COVID. Nervenarzt. 2022 Aug;93(8):779-787. German. doi: 10.1007/s00115-022-01297-z. Epub 2022 May 16. PMID: 35576015
4. Bocek J et. al., Sleep Disturbance and Immunological Consequences of COVID-19. Patient Prefer Adherence. 2023;17:667-677 https://doi.org/10.2147/PPA.S398188
5. Pang JCY et. al., The impacts of physical activity on psychological and behavioral problems, and changes in physical activity, sleep and quality of life during the COVID-19 pandemic in preschoolers, children, and adolescents: A systematic review and meta-analysis. Frontiers in Pediatrics. 11. 2023. doi.org/10.3389/fped.2023.1015943
6. Tedjasukmana R et. al., Sleep disturbance in post COVID-19 conditions: Prevalence and quality of life. Frontiers in Neurology. 13. 2023. doi.org/10.3389/fneur.2022.1095606  

Overall, this review manuscript is comprehensive and well-structured and has potential relevance especially during the ongoing COVID-19 related mental health issues.

Author Response

More than 11-12 citations have been repeated at least twice in the reference list. This should be corrected with proper referencing tool.

All the repeated citations including 6, 22, 70, 13, 14, 17, 52,64, 66, 75,93, 96, 98,100 have been removed and appropriated numbering was changed in article and references section.

At line 63-66, Authors have indicated about the lack of consistency between study results and to discuss these differences as one of the goals of this review. However, except few places such as at line 180, it is not apparent elsewhere in the current manuscript. Please briefly discuss or summarize those including the information based upon the listed citations 20-22.

More information regarding inconsistency in knowledge has been added.

The manuscript could greatly benefit from additional and updated information from few of the recent publications. The latest citation used in this manuscript is almost a year old (April-May 2022). Few of relevant recent publications are listed below for authors consideration: new updated references added
1.Scarpelli S et. al., 2022. Subjective sleep alterations in healthy subjects worldwide during COVID-19 pandemic: A systematic review, meta-analysis and meta-regression, Sleep Medicine, 100, 89-102, https://doi.org/10.1016/j.sleep.2022.07.012. Discusses- Sleep quality in different world areas, characterization of subjective sleep alterations during the COVID-19 pandemic, identification of significant predictors of sleep problems:

added

2.Bezner TL, Sivaraman M. COVID-19 Pandemic and Its Impact on Sleep Health: A Rapid Review. Mo Med. 2022 Jul-Aug;119(4):385-389. PMID: 36118813; PMCID: PMC9462915.
Discusses- Direct Impact of COVID on Sleep Health and the Unfortunate Synergy with OSA

added

3. Schilling C et. al., Cognitive disorders and sleep disturbances in long COVID. Nervenarzt. 2022 Aug;93(8):779-787. German. doi: 10.1007/s00115-022-01297-z. Epub 2022 May 16. PMID: 35576015

added

4. Bocek J et. al., Sleep Disturbance and Immunological Consequences of COVID-19. Patient Prefer Adherence. 2023;17:667-677 https://doi.org/10.2147/PPA.S398188

added

5. Pang JCY et. al., The impacts of physical activity on psychological and behavioral problems, and changes in physical activity, sleep and quality of life during the COVID-19 pandemic in preschoolers, children, and adolescents: A systematic review and meta-analysis. Frontiers in Pediatrics. 11. 2023. doi.org/10.3389/fped.2023.1015943

added

6. Tedjasukmana R et. al., Sleep disturbance in post COVID-19 conditions: Prevalence and quality of life. Frontiers in Neurology. 13. 2023. doi.org/10.3389/fneur.2022.1095606  

added

Reviewer 2 Report

This manuscript deals with "COVID-19 and sleep disturbances: A literature review of clinical evidence" I suggest a minor correction and require a detailed clarification. A correction should be addressed by the authors as follows: The abstract is not well organized; the sentences are incomplete, and there is no sense of continuity. It would be feasible if you included the significance of the current study in the abstract. A brief description of how the authors selected information from the literature in the databases, as well as what time period they searched for, is missing. The authors should justify and expand the information on the advantages of this work for biomedical applications. Authors should specify the main experimental conditions used based on the evidence from the literature. Where they briefly describe the most important data reported in the literature in a homogeneous manner and reinforce the relevance of natural compounds and nanotechnologies as novel alternatives. Authors should discuss whether the use of novel methods  represents a solid alternative to existing therapeutics. Please add the below studies to your manuscript in the discussion section and bold your study novelties: 

DOI: 10.2147/PPA.S398188

DOI:10.3390/microorganisms9020232

DOI:10.1016/j.genhosppsych.2023.01.002

Author Response

This manuscript deals with "COVID-19 and sleep disturbances: A literature review of clinical evidence" I suggest a minor correction and require a detailed clarification. A correction should be addressed by the authors as follows:

The abstract is not well organized; the sentences are incomplete, and there is no sense of continuity. It would be feasible if you included the significance of the current study in the abstract.
Rewritten abstract

A brief description of how the authors selected information from the literature in the databases, as well as what time period they searched for, is missing.

We appreciate the reviewer’s input, but since this is not a meta analysis or systemic analysis review, the Prisma/related guidelines are not applicable.

The authors should justify and expand the information on the advantages of this work for biomedical applications. Authors should specify the main experimental conditions used based on the evidence from the literature. Where they briefly describe the most important data reported in the literature in a homogeneous manner and reinforce the relevance of natural compounds and nanotechnologies as novel alternatives. Authors should discuss whether the use of novel methods represents a solid alternative to existing therapeutics.

We appreciate the reviewers input in this regard.  This emerging area falls beyond the scope of this manuscript, but agree with the importance of this modalities, we have added those brief discussion in the manuscript. Figure 2 summarizes the potential opportunities for nanotechnology use in COVID related pathologies. Also discussed the newer therapies we mentioned vs other “classic” therapies offered for sleep disorders/disturbances.

Please add the below studies to your manuscript in the discussion section and bold your study novelties: DOI: 10.2147/PPA.S398188:

added

DOI:10.3390/microorganisms9020232

This article is “A Comprehensive Review of Detection Methods for SARS-CoV-2”: I don’t feel this relevant to our article.

DOI:10.1016/j.genhosppsych.2023.01.002:

added

Figure 2: Nanotechnology uses in COVID-19 pandemic